METHODS AND RESOURCES

# iPHoP: An integrated machine learning framework to maximize host prediction for metagenome-derived viruses of archaea and bacteria

**Simon Roux** [1]*, **Antonio Pedro Camargo**[1], **Felipe H. Coutinho**[2], **Shareef M. Dabdoub**[3], **Bas E. Dutilh**[4,5], **Stephen Nayfach**[1], **Andrew Tritt**[6]

**1** DOE Joint Genome Institute, Lawrence Berkeley National Laboratory, Berkeley, California, United States of America, **2** Instituto de Ciencias del Mar (ICM-CSIC), Barcelona, Spain, **3** Division of Biostatistics and Computational Biology, University of Iowa College of Dentistry, Iowa City, Iowa, United States of America, **4** Institute of Biodiversity, Faculty of Biological Sciences, Cluster of Excellence Balance of the Microverse, Friedrich Schiller University, Jena, Germany, **5** Theoretical Biology and Bioinformatics, Science for Life, Utrecht University, Utrecht, the Netherlands, **6** Applied Mathematics and Computational Research Division, Lawrence Berkeley National Laboratory, Berkeley, California, United States of America

* sroux@lbl.gov

## Abstract

The extraordinary diversity of viruses infecting bacteria and archaea is now primarily studied through metagenomics. While metagenomes enable high-throughput exploration of the viral sequence space, metagenome-derived sequences lack key information compared to isolated viruses, in particular host association. Different computational approaches are available to predict the host(s) of uncultivated viruses based on their genome sequences, but thus far individual approaches are limited either in precision or in recall, i.e., for a number of viruses they yield erroneous predictions or no prediction at all. Here, we describe iPHoP, a two-step framework that integrates multiple methods to reliably predict host taxonomy at the genus rank for a broad range of viruses infecting bacteria and archaea, while retaining a low false discovery rate. Based on a large dataset of metagenome-derived virus genomes from the IMG/VR database, we illustrate how iPHoP can provide extensive host prediction and guide further characterization of uncultivated viruses.

## Introduction

Viruses are widespread and influential throughout all ecosystems. In microbial communities, viral infections can shift community composition and structure via viral lysis and alter biogeochemical processes and metabolic outputs through reprogramming of host cells during infection [1–3]. Given current challenges for cultivating many environmental microbes and their viruses, the extensive viral diversity is now primarily explored via metagenomics, i.e., by assembling genomes of uncultivated viruses directly from whole community shotgun sequencing data [4–6]. Over the last decade, metagenomic studies have been incredibly powerful in revealing viral diversity on Earth, investigating eco-evolutionary drivers of viral biogeography,

**Data Availability Statement:** All data directly relevant are within the paper and its Supporting Information files. Benchmarking analysis were based on the publicly available IMG/VR v3 database (doi 10.1093/nar/gkaa946 - https://genome.jgi.doe.gov/portal/IMG_VR/IMG_VR.home.html).

**Funding:** BED was supported by the European Research Council (ERC) Consolidator grant 865694: DiversiPHI, the Deutsche Forschungsgemeinschaft (DFG, German Research Foundation) under Germany's Excellence Strategy – EXC 2051 – Project-ID 390713860, the Alexander von Humboldt Foundation in the context of an Alexander von Humboldt Professorship funded by the German Federal Ministry of Education and Research, and the European Union's Horizon 2020 research and innovation program, under the Marie Skłodowska-Curie Actions Innovative Training Networks grant agreement no. 955974 (VIROINF). FHC was supported by a Juan de la Cierva - Incoporación fellowship (Grant IJC2019-039859-I), and had the institutional support of the "Severo Ochoa Centre of Excellence" accreditation (CEX2019-000928-S). This work was supported by the U.S. Department of Energy, Office of Science, Biological and Environmental Research, Early Career Research Program (SR) awarded under UC-DOE Prime Contract DE-AC02-05CH11231. The work conducted by the U.S. Department of Energy Joint Genome Institute (https://ror.org/04xm1d337), a DOE Office of Science User Facility, is supported by the Office of Science of the U.S. Department of Energy operated under Contract No. DE-AC02-05CH11231 (SR, APC, SN). The funders had no role in study design, data collection and analysis, decision to publish, or preparation of the manuscript.

**Competing interests:** The authors have declared that no competing interests exist.

**Abbreviations:** AAI, amino acid identity; CWF, complexity estimation by Wootton–Federhen; FDR, false discovery rate; GEM, Genomes from Earth's Microbiomes; LCA, lowest common ancestor; MAG, metagenome-assembled genome; PPV, positive predictive value; SAG, single-cell amplified genome.

and connecting viruses to ecological and metabolic processes [7–9]. A major limitation of these approaches is that metagenome-derived viral genomes have no inherent link with a host, as is the case for isolates [4]. This lack of host association remains a critical hurdle when attempting to study virus–host interactions and dynamics in natural communities, in particular for the highly diverse bacteriophages ("phages"), viruses infecting bacteria [4].

Given the importance of phage–host interactions in microbiome processes, computational methods to predict the host(s) of a phage based on its genome sequence are highly desirable, and the subject of active research [10,11]. Existing host prediction tools either leverage various levels and patterns of sequence similarity between phage and host genomes ("host-based" tools hereafter) or use a "guilt-by-association" approach by comparing the query phage to a database of viruses with known host(s) ("phage-based" tools).

In host-based tools, sequence similarity between phage and host genomes can be detected through sequence alignment, reflecting, e.g., prophages integrated in the host genome or similarity between phage genomes and host CRISPR spacers [10,12]. Alternatively, host-based tools can rely on alignment-free approaches, e.g., comparison of nucleotide k-mer frequencies, in which case these typically reflect the overall adaptation of virus genomes to their host cell machinery [13–22]. Because they rely on different signals, these host-based tools display varying levels of recall and specificity and are likely to be each relevant for different types of samples and viruses [10]. In previous benchmarks, alignment-based methods could reach high specificity when using strict cutoffs, e.g., >75% of predictions correct at the species level, but only for a limited subset of the input phages due to limitations of the host reference database [10]. Meanwhile, in the same benchmark, alignment-free methods appeared to contain a genuine and strong phage–host signal for a broader range of phages, but more complex to parse as the highest scoring host was often (>50% of the time) yielding an incorrect prediction at the species, genus, and family level.

In contrast, "phage-based" tools rely not on phage–host similarity, but extract information from a database of reference phages and archaeoviruses with known host(s) [23–27]. The most recent tools in this category have been the most promising overall, with benchmarks suggesting both high recall and high specificity. For example, RaFAH achieved a 33% improvement in F1 score (combination of recall and precision) at the genus level compared to host-based methods [25]. While phage-based approaches are particularly suitable if related phages exist with known hosts, RaFAH also predicted hundreds of archaeal viruses, i.e., domain-level host predictions, despite archaeoviruses being underrepresented in the database [25]. However, it remains unclear to what extent phage-based tools can provide reliable host prediction at lower ranks such as genus or species for entirely novel phages, and how to best complement these phage-based predictions with host-based signals [11].

With multiple tools available for host prediction, several studies have attempted to integrate the results from several approaches into a single prediction for each virus. This integration step was originally performed via empirical "rule sets" prioritizing methods based on empirical accuracy or error rate estimations [28,29]. Recently, several automated tools were developed that instead leverage machine learning to obtain an integrated host prediction. PhisDetector [30] combines multiple host-based methods, both alignment-based and alignment-free, and uses an ensemble of machine learning approaches to evaluate the confidence of each potential phage–host pair. VirHostMatcher-Net [31] proposes to integrate both virus–virus and virus–host signal in a modeled virus–host network, from which potential virus–host pairs are evaluated using a logistic regression. While both tools showed potential improvements compared to single methods, none of the benchmarks provided suggested that they could reach a low (<10%) false discovery rate (FDR) at the host genus level, even with the strictest cutoffs. In addition, no benchmark was carried out across different degrees of phage "novelty," i.e.,

different degrees of similarity to the most closely related reference, so it remains unclear how these approaches perform on "known" and "novel" phages.

Here, we present iPHoP, a tool for integrated Phage-Host Prediction, enabling high recall and low FDR at the host genus level for both known and novel phages. We first demonstrate the complementarity of phage-based and host-based approaches and describe a new modular machine learning framework that yields highly accurate predictions at the genus level. Using a diverse set of 216,015 metagenome-derived phage genomes, we further show that iPHoP enables high-confidence host genus prediction (estimated <10% FDR) for phages across a broad range of ecosystems and novelty compared to isolated references. iPHoP is available at https://bitbucket.org/srouxjgi/iphop, through a Bioconda recipe, and a Docker container.

## Results

To design an integrated framework for host prediction, we first evaluated the performance and complementarity of 10 existing methods on a common benchmark dataset [10,12–14,16,24–27]. We especially focused on comparing tool performances across a range of "novelty," i.e., using a test set that included both viruses closely related to references and viruses entirely novel.

### Limitations and complementarity of individual host prediction methods

A set of published alignment-based and alignment-free methods, either phage-based or host-based, was selected for benchmarking (S1 Table). These tools were evaluated on a common test dataset including bacteriophage and archaeovirus genomes available in NCBI GenBank but not included in NCBI RefSeq [32] and thus typically not used to train any of these tools (see Methods and S2 Table). This test dataset contained 1,870 genomes, spanning 170 host genera, including both temperate and virulent phages, and with both "known" and "novel" genomes (>90% and <5% genome-wide average amino acid identity, or AAI [33], to the closest reference, respectively; see S1 Fig). As host references, we opted to use all genomes included in the GTDB database [34], supplemented by additional publicly available genomes from the IMG isolate database [35] and the GEM catalog [36]. For each tool, we assessed host predictions at the host genus rank based on a naive "best hit" approach and using relaxed cutoffs (see Methods and S3 Table for other ranks).

First, we evaluated the recall of each tool, i.e., the total number of correct predictions obtained (Fig 1A). The recall differed across the tool categories, with the lowest observed for host-based alignment-based tools such as blast and CRISPR, and the highest observed for phage-based tools. The only exception was a very high recall observed for blast-based predictions of temperate phages, which is due to the detection of integrated copies of these phages, or closely related ones, in host genomes. A similar trend, i.e., a higher recall for temperate phages compared to virulent phages, was observed for most approaches albeit to a much lower degree (S2 Fig). For CRISPR-based predictions, the low recall compared to other approaches is likely due to limitations of the host database as CRISPR arrays can be absent from large clades of bacteria [37] and, when present, CRISPR spacers are typically highly variable even between closely related strains [10,38].

Next, we evaluated the precision of each tool, i.e., its ability to distinguish correct from incorrect hosts among all its predictions. As previously noted [10], host-based alignment-free tools struggled to achieve a high positive predictive value (PPV), i.e., a low FDR, even when using strict cutoffs (Fig 1B). In contrast, alignment-based tools, both phage-based and host-based, were able to reach high (>80%) PPV when filtering hits based on score(s). Pragmatically, this means that the scores provided by alignment-based tools are able to distinguish

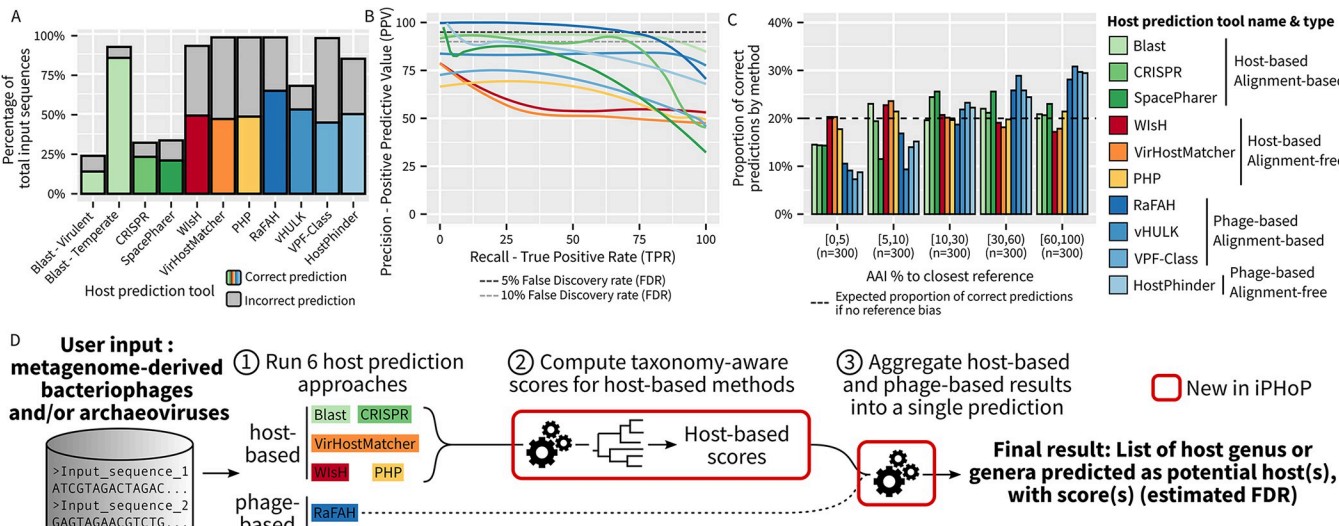

**Fig 1. Comparison of different host prediction approaches on a single test dataset.** (**A**) Total number of predictions and number of correct predictions (y-axis) obtained for each tool (x-axis) using a "best hit" approach and relaxed cutoffs (see Methods) on sequences from the test dataset (S2 Table). For each tool, the number of correct predictions is indicated by the colored bar, while the total number of predictions is indicated by the gray bar. Similar plots including the whole test dataset, virulent phages only, and temperate phages only are available in S2 Fig. (**B**) Precision-Recall curves for the different tools, using the same color code as in panels **A** and **C**. Two standard thresholds, 5% and 10% false discovery rates, are indicated by horizontal dashed lines. (**C**) Relationship between "novelty" of input virus, represented as AAI (average amino acid identity) percentage to the closest NCBI RefSeq reference on the x-axis, and the number of correct host predictions obtained with each tool. To evenly represent both "known" and "novel" input viruses, 300 sequences were randomly subsampled from each AAI percentage category (x-axis). (**D**) Schematic overview of iPHoP host prediction pipeline. Detailed explanations of the new steps 2 and 3 are available in Figs 2 and 3, and source data in S1 Data.

correct from incorrect predictions, while the scores provided by alignment-free tools are usually not sufficient to identify correct predictions.

Phage-based tools thus seemingly present an ideal combination of high recall and high precision, with RaFAH [25] in particular able to maintain a very low FDR (<5%) while providing the highest recall of all tools (Figs 1 and S2). However, phage-based tools depend on the availability of a related phage with a known host in the reference database. Specifically, phage-based tools mostly provide predictions for phages that are related to reference sequences and much less frequently for "novel" phages (<5% AAI to closest reference; Fig 1C). This reference bias was observed for all phage-based tools, including the one based on k-mer similarity (HostPhinder; Fig 1C). A similar trend, although less pronounced, can be observed for host-based tools relying on sequence alignment. Meanwhile, alignment-free host-based tools show little to no bias for phages with closely related references, suggesting that these methods would be well suited for dealing with the most "novel" phages. This bias is important to consider because the vast majority (57% to 79%) of viral genomes identified from metagenomes have <5% AAI to their closest reference (S3 Fig), so that phage-based approaches alone would not yield reliable host predictions.

Ultimately, these simplified benchmarks suggest that to tackle diverse "known" and "novel" phages, as typically obtained through metagenomics, host prediction tools will need to combine phage-based and host-based approaches. For phage-based approaches, several tools such as RaFAH already provide both high recall and high precision. Conversely, all current host-based methods suffer from either limited recall (alignment-based methods) or limited specificity (alignment-free methods), at least when used individually and in a simple "best hit" approach. In addition, the predictions from different host-based methods partially overlap, suggesting that multiple methods could be considered together to either reinforce or correct each other (S4 Fig). For our integrated host prediction tool, we thus decided to first optimize host-based predictions by

integrating multiple hits per method and several methods together, and then combine these host-based predictions with an established phage-based method to derive a single host prediction.

## Increasing host prediction accuracy by robustly integrating multiple hits for each virus

Elevated FDRs with host-based methods have been highlighted previously [10,13,14]. Traditionally, these have been addressed by applying relatively strict cutoffs on the prediction score, and by considering an arbitrary number of hits passing these cutoffs, e.g., the 5 or 10 best hits. These hits might be further integrated using a lowest common ancestor (LCA) approach. Intuitively, this will allow to distinguish reliable cases, where the top hits all point to the same host taxon, from unreliable cases where the top hits correspond to different taxa and the host predictions only reach consensus at a higher rank. Alternatives to LCA approaches have been proposed including the taxonomy-aware sequence similarity ranking framework from PHIRBO [39]. Here, we explored whether machine learning approaches could help improve these predictions by integrating all hits obtained for a virus using a given method.

To consider an ensemble of hits in a taxonomy-aware context, we opted to treat each hit as a separate classification problem, i.e., "is this host hit reliable or not considering the context of other hits obtained for this same virus with the same approach?". For a given input genome, each hit is thus considered as a candidate host, and an ensemble of hits for a virus is provided as input to different classifiers with information on the hits quality as well as phylogenetic distances between each hit and the candidate host based on the GTDB [34] framework (Figs 2A and S5). The task asked of the classifiers is to predict whether the candidate host belongs to the correct host genus, and the underlying assumption is that classifiers would learn to recognize reliable series of hits, e.g., cases where most of the top hits are close to the candidate host, from unreliable series of hits, e.g., cases where hits are distributed across diverse hosts and/or distant from the candidate host, without having to resort to arbitrary cutoffs (Fig 2B).

To evaluate this approach, we applied it separately to 5 host-based methods (Blast, CRISPR, WIsH [13], VirHostMatcher [14,31], and PHP [16]; see S1 Table), used RefSeq Virus sequences to train and optimize 3 types of classifier, namely dense neural networks, convolutional neural networks, and random forest classifiers (S4 Table), and compared the results obtained on the test dataset (see above) to a standard best hit approach (S5 Fig). Overall, considering multiple hits with automated classifiers reduced the error rate (average FDR) for all methods and all types of classifiers, with the highest reductions obtained with convolutional neural networks (Figs 2C and S6). This reduction in average error rate was especially important for WIsH- and CRISPR-based predictions (>40%), and smaller for BLAST, for which standard scores already seem to perform well.

Finally, we verified whether different variants of each classifier could be complementary, i.e., provide reliable scores for different types of sequence. In all cases, a set of 2 variants appeared to be the best combination to maximize the number of correct predictions while minimizing the FDR (see Methods). The 10 classifiers that were ultimately selected (2 for each of the 5 host-based methods) showed improved PPV, often >75%, at most true positive rates, confirming their improved ability to distinguish likely and unlikely candidate hosts compared to the raw score of each method (Fig 2D).

## Integrating host- and phage-based predictions for a comprehensive coverage of phage diversity

After optimizing the scoring systems for each host-based method, the next step was to integrate predictions across different methods to obtain a single prediction score taking into

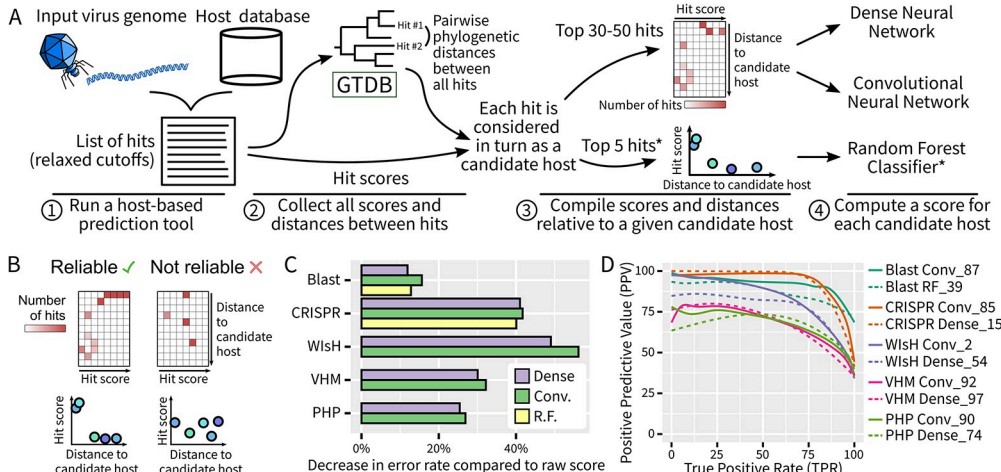

**Fig 2. Overview of the single-tool classifiers used in iPHoP.** (**A**) Schematic representation of the process used to score individual hits from host-based tools. Briefly, each hit was scored by a neural network or random forest classifier, which also considered other top hits for the same virus and the same tool. This process was applied to the 5 host-based tools selected ("Blast," "CRISPR," "WIsH," "VHM," "PHP"), except for the random forest classifiers (highlighted with a *), which were only used for "Blast" and "CRISPR." When considering multiple hits, their similarity or difference in terms of host prediction was estimated from the GTDB phylogenies [34]. (**B**) Illustration of how multiple hits are represented in neural networks input matrices (top) or random forest classifier inputs (bottom). Two examples are provided, one "reliable" in which the hits with high scores are all consistent and at a small distance to the candidate host considered (left), and the other "unreliable" in which a few hits with medium-to-high scores are scattered across hosts with variable distance to the candidate host considered. (**C**) Estimated improvement in classification provided by the automated classifiers compared to "naive" raw scores. These estimations are based on smoothed ROC curves obtained from the test dataset (see S6 Fig) and calculated as the average decrease in false discovery rate for 17 true positive rates ranging from 10% to 90%. Random forest classifiers were only evaluated for Blast and CRISPR approaches. (**D**) Precision Recall curves for the 2 classifiers selected for each host-based tool (see S4 Table). Conv, "Convolutional Neural Network"; "RF", "random forest classifier"; VHM, "VirHostMatcher." Source data are available in S1 Data.

account all different approaches for each potential phage–host pair. Traditionally, this has been done using fixed "rule sets" informed by estimation of FDR for each approach, e.g., prioritizing alignment-based approaches over alignment-free approaches [28,29]. Here, we instead used a 2-step integration process to robustly consider all hits for each input sequence.

First, to leverage the high sensitivity of alignment-free approaches but reduce their error rate, we trained and optimized a random forest classifier based on the scores from the 10 individual host-based classifiers described in the previous section ("Combined-hosts-RF classifier"; Fig 3A). This Combined-hosts-RF score yielded low FDR ($\leq$10%) even at high TPR ($\geq$75%) and was comparable in that regard to the scores obtained from the phage-based tool RaFAH [25], as well as host-based aligment-based tools (Blast and CRISPR; Fig 3B).

Next, we designed a composite confidence score ("iPHoP score") for each phage–host pair to summarize results from both phage- and host-based methods (Fig 3A; see Methods). Because BLAST- and CRISPR-based predictions can also be reliable on their own without the need for any other approach (Fig 2D), we included the best score for each of these approaches along with the Combined-hosts-RF score and the score from RaFAH [25], the most reliable phage-based tool in our benchmark (Fig 1A and 1B). As expected based on our initial benchmarks (Fig 1C), different methods provided correct host predictions for different input phages, and combining them led to high rates of phages with correct predictions ($\geq$50%) for both "known" and "novel" phages (S7 Fig).

To illustrate the unique features and performance improvements provided by iPHoP, we compared it to other automated tools integrating multiple approaches for host prediction,

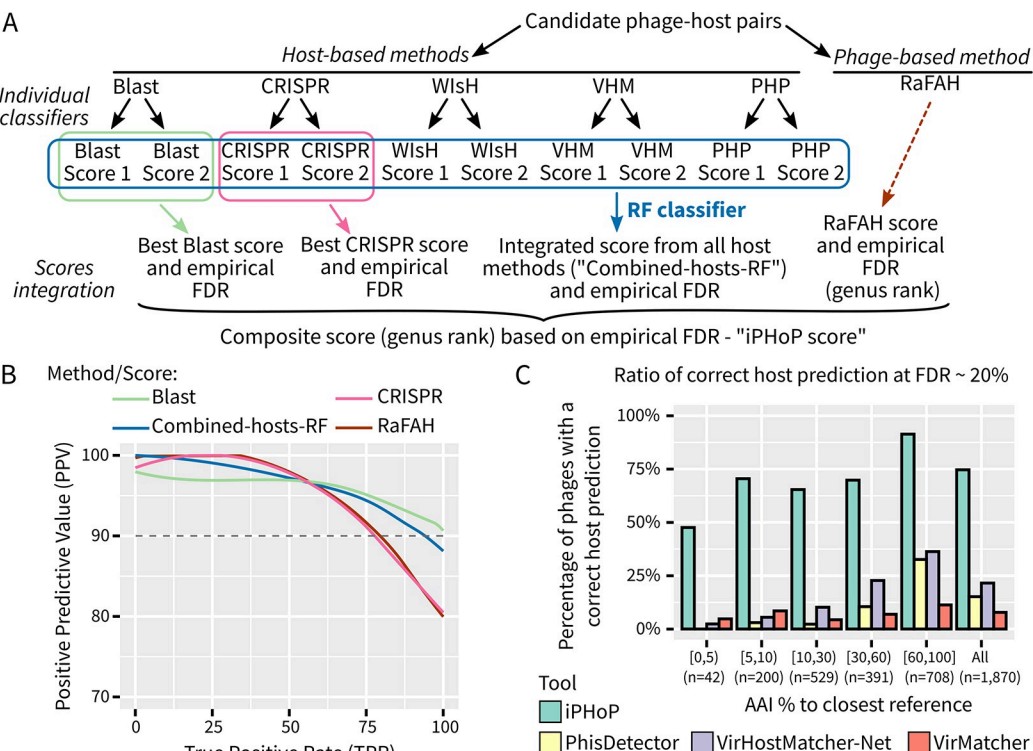

**Fig 3. Schematic and performance of iPHoP host genus predictions.** (**A**) Schematic representation of the integration process. "Individual classifiers" refer to single-tools scores calculated for each virus–candidate host pair (see Fig 2). (**B**) Precision Recall curve for each of the 4 scores considered in iPHoP composite score, based on the test dataset. (**C**) Comparison of the percentage of input sequences from the test dataset for which a correct host genus prediction was obtained, when using cutoffs limiting FDR to 20% maximum. This percentage is given for all sequences in the test dataset ("All"), and for subsets of sequences defined based on their amino acid similarity to the closest reference phage genome in the NCBI RefSeq or RaFAH database. AAI, amino acid identity; FDR, false discovery rate; RF, random forest. Source data are available in S1 Data.

namely VirMatcher [29], PhisDetector [30], and VirHostMatcher-Net [31]. Based on Receiver Operating Characteristic and Precision Recall curves, iPHoP performed as well as, or better than all other integrated tools (S8 Fig). However, the major improvement of iPHoP comes from the number of phages with a host prediction: for a given FDR, iPHoP typically provides approximately 3 to 5 times more predictions than the next best tool, especially for "novel" phages (Fig 3C). This is likely due to the fact that (i) iPHoP uniquely leverages both phage-based and host-based approaches; (ii) iPHoP integrates more approaches than any other tool; (iii) the iPHoP host database is larger and more diverse than those used by other tools; and (iv) iPHoP was specifically optimized for predictions at the host genus rank. In contrast, VirHost-Matcher-Net relies on a network architecture to represent virus–host interactions and derive host predictions at multiple taxonomic ranks, while PhisDetector was designed to provide host predictions down to the species rank [30,31].

## Expanding host predictions in a large database of metagenome-derived viruses

To further evaluate the improvements provided by iPHoP and the remaining challenges when analyzing diverse metagenome-derived phage genomes, we applied iPHoP to 216,015 high-quality (i.e., predicted to be ≥90% complete by CheckV) IMG/VR sequences (S3 Fig). We

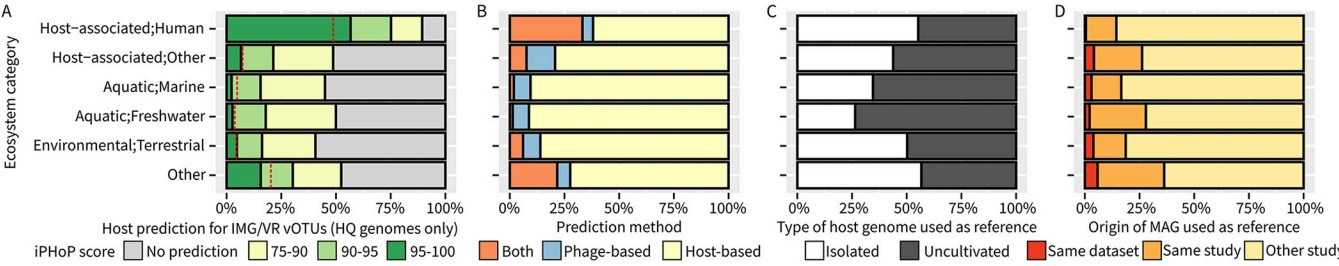

**Fig 4. Overview of iPHoP host prediction for high-quality IMG/VR v3 genomes.** (**A**) Distribution of the best iPHoP score for high-quality genomes from the IMG/VR v3 database by ecosystem. For each IMG/VR vOTU, the best score from iPHoP was considered if ≥75, or the vOTU was considered as not having a predicted host. The proportion of sequences for which a host prediction was available in the original IMG/VR database is indicated with a dashed red line. (**B**) Distribution of the type of signal used to achieve host prediction with a score ≥90 in iPHoP. "Host-based" includes all 5 host-based tools, while "Phage-based" includes predictions obtained with RaFAH. "Both" includes consistent predictions obtained with RaFAH and at least 1 host-based tool. (**C**) Percentage of hits from isolated or uncultivated host genomes used in host-based predictions with iPHoP scores ≥90. These are based on the individual genome hits underlying iPHoP genus-level predictions. (**D**) Origin of the uncultivated host genomes used in host-based predictions with iPHoP scores ≥90. The original dataset and study ID for the query virus and the uncultivated host genome were obtained from the Gold database, and when both were available, these were compared to evaluate whether the uncultivated host genome originated from the same dataset, a different dataset from the same study, or another study from the query virus. Source data are available in S1 Data.

then compared the iPHoP predictions to the current host predictions available in the IMG/VR database, which were primarily based on blast hits to host genomes and CRISPR spacers [8] (Fig 4A). Overall, iPHoP predictions at an estimated FDR ≤10%, i.e., iPHoP score ≥90, represented a 1.5- to 13-fold increase compared to the original number of host prediction in the IMG/VR v3 database; however, these numbers vary greatly depending on the ecosystem (Fig 4A). For human-associated microbiomes, about 89% of the nonredundant high-quality genomes had a host predicted using iPHoP, including 57% with very high confidence predictions (iPHoP score ≥95). For all other ecosystems, the total number of phages with predictions was lower, ranging from approximately 40% to 50%, including approximately 15% to 22% with medium or high confidence (iPHoP score ≥90). For comparison, VirHostMatcher-Net applied to the same dataset yielded a host prediction for 23% of the input sequences for a comparable estimated FDR of 15%, with the majority (86%) of these sequences also being connected to a predicted host via iPHoP. Across ecosystems, host predictions originated primarily from host-based methods, consistent with the high number of metagenome-derived sequences unrelated to those in the reference databases (Figs 4B, S3 and S9). Human microbiomes again stand out with >25% of host predictions confirmed by both phage- and host-based methods, which explains the high number of high-confidence predictions (Fig 4A). For all ecosystems, iPHoP provided host prediction for both temperate and virulent phages, although a higher percentage of predictions was obtained for temperate ones (S10 Fig). While these results reflect the inherent bias in current microbial and phage reference databases, they suggest that iPHoP is already useful across different biomes and for different virus types and may be expected to improve as more of the global microbial and viral diversity is characterized.

Within host-based approaches, nearly half (mean: 45%) of the predictions were based on genomes of uncultivated bacteria and archaea, highlighting the value of using large databases including single-cell amplified genomes (SAGs) and/or metagenome-assembled genomes (MAGs) [34,36] (Fig 4C). These genomes from uncultivated microbes were particularly important for predicting hosts of environmental phages, especially freshwater and marine phages (Fig 4C). This was confirmed by performing host prediction with iPHoP for the same sequences using a custom database without any MAG, which yielded 9.95% less host prediction on average, including 17.65% and 20.71% less for freshwater and marine phages, respectively (S5 Table). Given this increased prediction rate, public MAGs from uncultivated bacteria and archaea (approximately 60,000) were included in the iPHoP default host database.

We next wondered what proportion of these hosts were "local," i.e., assembled from the same sample as the query phage or another sample in the same study. Overall, in several ecosystems, a substantial (>25%) proportion of MAGs used for host predictions were obtained from metagenomes generated in the same study from which the input phage was derived (Fig 4D). We confirmed the relevance of "local" MAGs by performing host prediction with iPHoP for 3 specific studies using custom databases where host MAGs from these studies were removed, which resulted in 26% of the host predictions being lost (S5 Table). Hence, for comprehensive host prediction of a new phage dataset, it may be valuable to also integrate into the host genome database additional bacterial and archaeal MAGs obtained from the same sample or experiment, if available. To facilitate this, we included an automated database building module in iPHoP, enabling users to add their own MAGs in a host database based on phylogenies and taxonomies generated through GTDB-tk [40].

### Estimating host diversity coverage by metagenome-derived viruses

We next evaluated these IMG/VR host predictions from the host perspective, specifically assessing which host taxa were most frequently associated with viruses, and how much of the bacterial and archaeal diversity remained without any known or predicted virus. Overall, across the 5,711 bacteria and archaea genera with at least 2 genomes in the host database, 205 (3.6%) were associated with at least 1 reference virus in NCBI RefSeqVirus, while 1,700 (29.8%) were exclusively associated with metagenome-derived virus(es) through iPHoP (iPHoP score ≥90; Fig 5A and S6 Table). These host genera only associated with viruses through iPHoP predictions were found across various bacterial and archaeal phyla, from *Firmicutes* and *Bacteroidota* to *Methanobacteriota* (31% to 48% of genera with host prediction only; Fig 5A), and were not systematically associated with the largest genera, i.e., the ones with the highest number of species (S11 Fig). Meanwhile, other phyla such as *Patescibacteria*, *Planctomycetota*, *Acidobacteriota*, and *Chloroflexota* still displayed a majority of genera without any associated virus, either isolated or predicted (79% to 83%), highlighting the large diversity of viruses likely still to be identified and characterized.

We also evaluated which host taxa were associated with the largest number of predicted viruses for each biome reasoning that, if the predictions were mostly correct, these should correspond to taxa that are frequently observed in these ecosystems. Overall, the 10 genera most frequently predicted as hosts in each ecosystem indeed corresponded to taxa primarily detected in these same biomes, e.g., *Bacteroides* and *Faecalibacterium* for human microbiome, *Vibrio* and *Pseudoalteromonas* for marine samples, and *Streptomyces* and *Mycobacterium* for terrestrial samples (Fig 5B). The main exception to this pattern was the unexpectedly high number of host predictions to the *Bacteroides* genus for marine, freshwater, and terrestrial viruses. As the *Bacteroides*-infecting *Crassvirales* phages [41–43] have been used as markers for fecal contamination [44,45], these predictions might reflect pervasive contamination of these environments, although these may also reflect a bias in the current phage and host isolate databases, skewing predictions towards this host genus. Overall, while these results illustrate how in silico host predictions must always be considered critically and in light of the current limitations of databases and tools, increasing the diversity of isolated phage–host pairs from various environments will likely help refine these predictions in the future.

### Partial genomes, novel host genera, and eukaryotic viruses as potential sources of errors

Finally, we explored the impact on iPHoP prediction of 2 types of input sequences possibly occurring in metagenome analyses: partial genomes of bacteriophages and archaeoviruses, as

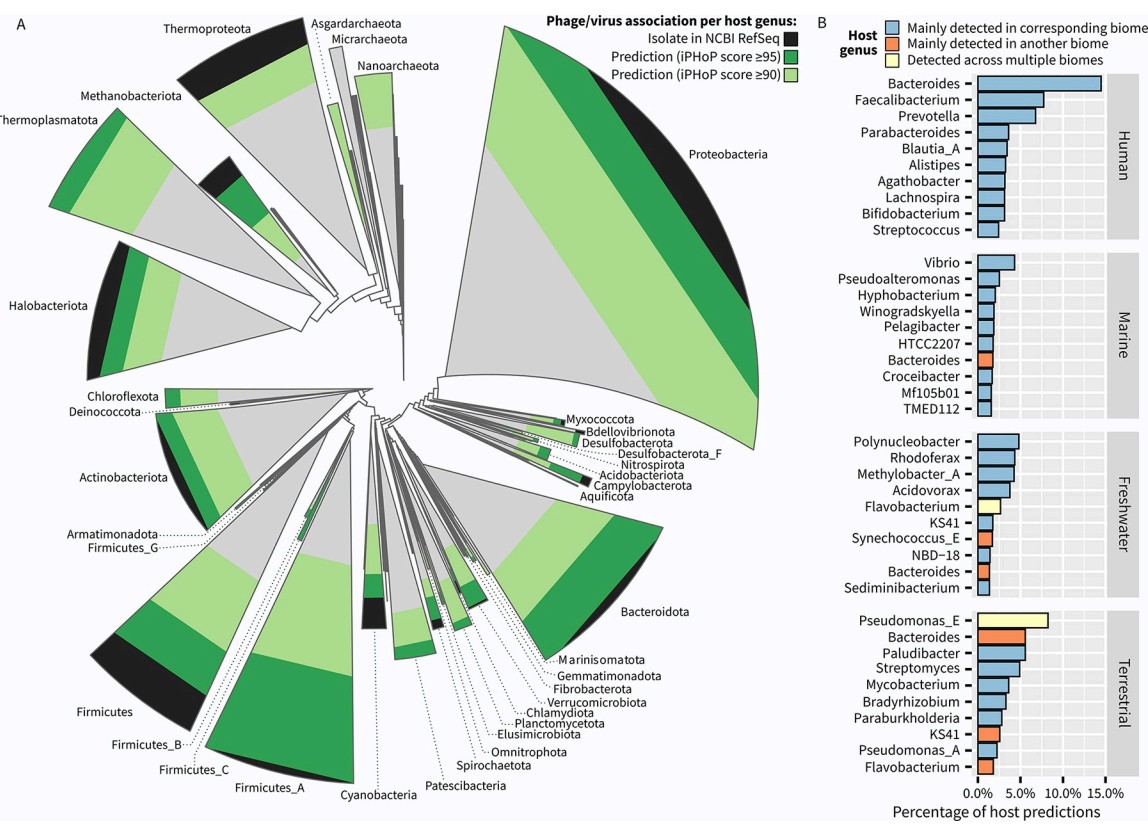

**Fig 5. Taxonomic and environmental distribution of hosts predicted using iPHoP from the IMG/VR v3 genomes.** (**A**) Archaeal (top left) and bacterial (bottom right) genome diversity from the GTDB database r202 [34]. The GTDB phylogenetic trees were collapsed at the phylum level. The status of virus association, i.e., isolated virus, predicted virus only at iPHoP score ≥95 or ≥90, or no prediction, was evaluated for each host genus, and the phyla shapes are colored according to the number of genera in each category within this phylum. (**B**) For each major biome type, the 10 host genera with the highest number of predicted IMG/VR high-quality virus genomes are included in the plot. Each host genus was also determined to be mainly detected in a biome type or detected across multiple biomes based on the distribution of MAGs assigned to this genus across ecosystems in the GEM catalog (see Methods). Source data are available in S1 Data.

opposed to near-complete genomes, and genomes of eukaryotic viruses. Despite recent progresses in sequencing technologies and bioinformatic tools, phage genome assembly can still yield mostly "low-quality" (<50% complete) genomes for some samples and ecosystems, which will likely negatively impact downstream host prediction. Meanwhile, some virus sequence identification tools can detect both eukaryotic and prokaryotic viruses, and these sequences are sometimes analyzed as part of the same dataset, which may lead to errors for host prediction tools like iPHoP that expect bacteriophages and archaeoviruses as input.

When applying iPHoP to randomly fragmented sequences from the test set, we observed a systematic decrease in recall with decreased length and/or completeness (S12 Fig). However, the FDR observed for these partial genomes was similar to that observed for complete genomes for "high" cutoffs (minimum iPHoP score 90 and 95), and only slightly increased at minimum iPHoP score 75 (from approximately 20% to approximately 25%). Taken together, this suggests that applying iPHoP to partial genomes is likely to result in reduced recall, i.e., fewer input sequences will receive a prediction with a high score, but the number of erroneous predictions is unlikely to increase substantially.

Next, we evaluated the results of iPHoP host prediction for phages infecting "novel" hosts, i.e., a host genus without a representative genome in the host database. To that end, we applied iPHoP on the test set using filtered databases in which all reference genomes from the real host

genus were removed (S13 Fig). Overall, when using minimum score cutoffs of 90 or 95, the majority of phages did not yield a prediction when the real host genus was filtered from the database, and when a prediction was provided, it was to a genus in the same family as the real host genus. For a more relaxed minimum score cutoff of 75, iPHoP provided a prediction for 67% of the input phages, including to genera in the same order or class as the real host (S13 Fig). This suggests that, when processing samples with a large number of novel host genera, i.e., not represented in the current reference genome databases, iPHoP predictions should ideally be restricted to "high" score (i.e., score ≥90) and interpreted at the family rank.

Finally, when applied to 8,128 eukaryotic virus genomes from RefSeq, iPHoP predicted a bacterial or archaeal host for 1,018 viruses (S7 Table). These erroneous predictions were primarily (85%) derived from the k-mer comparison to bacteria and archaea genomes, for relatively short viral genomes within the RNA viruses (*Riboviria* realm, 640; 63%) and ssDNA viruses (*Monodnaviria* realm, 155; 15%), and the vast majority (90%) were associated with a relatively low confidence score (iPHoP score <90). These results warn that false-positive predictions may occur when users apply iPHoP to a mixed dataset containing both prokaryotic and eukaryotic viruses, especially for samples composed of RNA and/or ssDNA virus sequences. To alleviate this potential issue, we recommend users to cross-reference any iPHoP prediction with a host domain prediction, i.e., a prediction of whether the sequence belongs to a prokaryotic or eukaryotic virus, based on taxonomic assignment of this virus or on an automated tool such as Host Taxon Predictor [46].

## Discussion

Viral metagenomics has profoundly transformed our understanding of global viral diversity and viral impacts on microbial communities. One critical piece of information missing compared to isolated viruses is the host connection, which significantly limits the inference and biological knowledge extracted from viromics data [4]. Accordingly, different methods have been developed to address this critical challenge, each with their specific limitation. Here, we present the iPHoP framework as a way to automatically integrate results from multiple host prediction approaches, which enables reliable prediction of host genus for a larger diversity of phages than any previous tool. The iPHoP tool and database are available as a stand-alone tool (bitbucket.org/srouxjgi/iphop/), a Bioconda recipe (https://bioconda.github.io/recipes/iphop/README.html), and a Docker container (https://hub.docker.com/r/simroux/iphop).

While iPHoP substantially improved host predictions on viruses from real metagenomic datasets, several limitations remain. First, because it relies on a suite of different tools, iPHoP remains relatively slow compared to other tools: A full iPHoP host prediction takes approximately 12 minutes for a test set of 5 complete phage genomes using the Sept_2021_pub database and 6 CPUs. This running time may not be problematic for viromics studies that typically run host prediction only once on a large set of metagenome-derived virus genomes, but it makes iPHoP suboptimal for time-sensitive analyses. Second, while iPHoP scores are designed to reflect FDRs, these estimations depend on the composition of the test dataset used. Even though we tried to use a balanced set as much as possible by ensuring that we included viruses with a broad range of relatedness to reference sequences, iPHoP scores should only be interpreted as approximated FDRs at best. Third, since iPHoP was designed with a viral ecology framework in mind, our goal was to provide reliable host predictions at the genus rank, i.e., with FDRs ideally <10%, from diverse input phages. Arguably, in other contexts such as phage therapy applications, host predictions will need to be more specific and reach the host species or strain level. Such a high-resolution host prediction will likely require the reconstruction of detailed virus–host networks, as attempted by VirHostMatcher-Net [31], or detailed analysis

of receptor-binding proteins [23]. In the near future, however, we anticipate that genus-level approaches like iPHoP will be broadly applicable and provide host predictions for a large range of viruses, while higher-resolution approaches such as VirHostMatcher-Net will likely be more limited in scope, so that both types of tools will be useful for different applications. Fourth, several potential improvements to iPHoP can already be envisioned, including, e.g., the addition of complementary approaches such as the detection of shared tRNA between phages and hosts, the consideration of additional features such as whether the input virus is temperate or virulent, and the ability to predict multiple hosts for broad host range phages. Finally, iPHoP remains limited by host, virus, and host–virus databases, as illustrated by the difference in the number of phages with host prediction between the human microbiome and other ecosystems. Achieving similar performance across all biomes will require in particular expanding the catalog of potential host genomes, with a particular attention paid to CRISPR arrays, which are often not fully assembled from metagenomes [28,47], and expanding the diversity of viruses associated with a host, either from isolation or using in vitro host linkage [48–51]. In that context, to accommodate future expansions of the tool set and databases, iPHoP was intentionally designed as a modular framework, and we envision the current tool as only the first step towards a comprehensive automated in silico host prediction toolkit.

## Material and methods

### Virus–host training sets and host databases

To evaluate different host prediction approaches and train new classifiers, a curated dataset of known virus genomes with corresponding host taxonomy was established based on genomes available in the NCBI databases up to January 2021. For training new classifiers, sequences from bacteriophages and archaeoviruses were obtained from NCBI RefSeq release 201 (released in July 2020), and the host genus of each virus was obtained from the corresponding genome annotation and/or publication [52]. This dataset was used to train new classifiers (see below), but not to evaluate any tool since these virus–host pairs were likely to have been used for training in previously published tools as well.

To complement this training set, a distinct test set was established based on NCBI GenBank. Specifically, INPHARED [32] was used to download a collection of bacteriophages and archaeoviruses from NCBI in January 2021, and all genomes already present in NCBI RefSeq release 201 were removed. For the remaining ones, host taxonomic information was obtained from the corresponding annotation and/or publication, and genomes for which host taxonomy was uncertain were removed, leading to a final dataset of 1,870 viruses with host taxonomy (S2 Table). These genomes were compared to the NCBI RefSeq references (see above) as well as the phage reference database used in RaFAH v0.1 [25] using diamond blastp v0.9.24 (default parameters; [53]) after de novo prediction of cds using Prodigal v2.6.3 (option "-p meta"; [54]), and the AAI estimation script provided with the Metagenomic Gut Virus catalogue (https://github.com/snayfach/MGV/blob/master/aai_cluster; [33]). Temperate phages were identified in the test set based on the annotation provided with each genome by searching for the keywords "prophage," "provirus," "lysogen," and "integrated" and based on BACPHLIP v0.9.6 [55] with a minimum score of ≥0.8. When annotation and BACPHLIP prediction were conflicting, the information from the genome annotation was prioritized. Virulent phages were identified based on BACPHLIP v0.9.6 [55] with a minimum score of ≥0.8.

### Host database consolidation

The host genome database currently used in iPHoP, named "iPHoP_db_Sept21," was built from 3 publicly available genome sets, namely the GTDB database (release 202; [34]),

published genomes from the IMG database (as of July 7, 2021; [35]), and the Genomes from Earth's Microbiomes (GEM) catalog [36], as follows. First, the 47,894 representative genomes from each GTDB species cluster were obtained from the GTDB database itself. Next, bacteria and archaea genomes from the IMG database that were not already included in GTDB release 202 and with a total length ≥100 kb ($n$ = 22,188), and medium- and high-quality MAGs from the GEM catalogue ($n$ = 52,515), were compared to the GTDB species representatives using the ani_rep function from GTDB-tk v1.5.0 (default parameters; [40]), based on Mash version 2.3 [56] and FastANI v1.32 [57]. All genomes with a similarity of ≥99% ANI over ≥99% AF were considered as identical to one of the GTDB representatives and discarded ($n$ = 1,724). Nonidentical genomes with a similarity of ≥95% ANI to one of the GTDB representatives were retained in the database as members of the corresponding species cluster ($n$ = 32,735). Finally, the remaining genomes ($n$ = 27,279) were considered as potential representatives of additional species clusters. To include these in a GTDB-compatible phylogenomic framework, these genomes were first checked for quality using CheckM v1.1.3 [58], discarding all genomes with <50% completeness or >10% contamination, and then dereplicated with dRep v3.2.2 [59] with cutoffs of 90% ANI and 60% coverage. The nonredundant genomes ($n$ = 13,658) were then integrated in updated bacteria and archaea phylogenomic trees using the function de_novo_wf from GTDB-tk v1.5.0 (default parameters; [40]). The resulting trees are then used in iPHoP for taxonomic assignation of and phylogenetic distance estimation between all representatives.

The representative genomes included in the GTDB-tk-generated trees are used in iPHoP for all prediction methods ($n$ = 60,000, i.e., 47,894 existing GTDB representatives and 12,106 additional ones from IMG and GEM). For blast-based prediction, these representative genomes are supplemented with additional genomes clustered into one of these species clusters, after removing duplicate genomes ($n$ = 43,022, for a total of 103,022 genomes used). Finally, for CRISPR-based predictions, CRISPR spacers were predicted de novo in all 121,781 genomes (i.e., representatives, clustered, and duplicates), with CRT 1.2 [60] and PilerCR [61] using custom python scripts from https://github.com/snayfach/MGV/tree/master/crispr_spacers [33]. All spacer sequences from arrays with ≥3 spacers were collected and dereplicated (100% identity), and spacers with a sequence length <10 or >100 were excluded. Ultimately, the spacer collection used in the iPHoP_db_Sept21 database includes 1,398,130 spacers from 40,036 distinct genomes.

## Evaluation and benchmarking of selected host prediction methods

A set of published tools performing host predictions based on a single approach were selected for benchmarking and potential inclusion in the iPHoP integrated framework (S1 Table). All these tools were benchmarked against the same test dataset (see above) established from virus sequences from NCBI GenBank (January 2021). Blast-based predictions were based on a blastn comparison (v2.12.0+, maximum e-value $1 \times 10^{-3}$, minimum identity percentage 80, maximum target sequence 25,000, minimum hit length 500 nt; [62]) between the input virus genomes and the iPHoP_db_Sept21 blast database (see above). Metrics considered for each pair of input virus and host contigs were total number of matches and average identity percentage. CRISPR-based predictions were based on a blastn comparison (v2.12.0+, word size 7, no filtering of hits based on low complexity, i.e., "-dust no," maximum target sequence 10,000,000; [62]) between the input virus genomes and the iPHoP_db_Sept21 spacer database (see above), considering only hits to spacers 25 nucleotides or longer, with less than 8 mismatches overall, and with a custom complexity score <0.6. The custom spacer complexity score was calculated based on sequence AT skew content and the complexity estimation by

Wootton–Federhen (CWF) [63] as follows: The complexity score is set as (CWF score– 2) * 2, except if (AT skew) > 0.65, in which case the complexity score is set to (AT skew) + 0.1. For Figs 1A and S2, only hits with 2 or less mismatches over the entire spacer were considered. The metric considered to rank hits for individual input virus genomes was the total number of mismatches when considering the entire spacer. SpacePHARER predictions were based on the predictmatch function from SpacePHARER v2.fc5e668 [12] applied to the input virus genomes with a sensitivity of 7.5 ("-s 7.5") and a maximum number of results per query sequence of 10,000, using the IMG/VR v3 CRISPR database [8]. SpacePHARER "Combined score" metric was used to rank predicted hosts for each input, with a minimum score cutoff of 20 applied for Figs 1A and S2.

For WIsH predictions, input virus genomes were compared to the iPHoP_db_Sept21 WIsH database with WIsH v1.0 [13] and a maximum $p$-value of 0.2. The WIsH $p$-value was also used to rank predictions for each input virus. For predictions based on the $s_2^*$ similarity, the corresponding code from VirHostMatcher-Net (July 2021 version; [31]) was used to compare input virus genomes to the VirHostMatcher database in iPHoP_db_Sept21 (see above). The $s_2^*$ similarity score is the only metrics considered for each hit. For PHP, input virus genomes were compared to the iPHoP_db_Sept21 PHP database (see above) using PHP (July 2021 version; [16]), and the PHP score was used as a metric for each hit. The upset plot comparing the predictions obtained for different host-based tools was generated with the UpSetR package [64].

RaFAH [25] predictions were obtained by running the "predict" function from RaFAH v0.3 on the input virus genomes with default parameters and using the Predicted_Host_Score as metric. vHULK [26] predictions were obtained by running vHULK v1.0.0 with default parameters and using score_genus_relu as the metric. VPF-Class predictions were obtained by running the vpf-class function from the vpf-tools 0.1.0.0 toolkit [27] with default parameters, using the host taxon with the highest membership_ratio as the prediction for each input virus and the confidence_score as the metric. Finally, HostPhinder predictions were obtained by running the "latest" version of HostPhinder docker container (December 2015) with default parameters and the main reported score as metric.

For all the tools, the best prediction was taken for each input virus based on the relevant metric and considered as correct if the genus of the predicted host genome or the predicted genus for tools predicting host taxonomy was consistent with the information collected from the reference database (S2 Table).

## Establishment of balanced training sets for single-tool iPHoP classifiers

For the 5 host-based approaches selected to be included in the iPHoP framework ("Blast," "CRISPR," "WIsH," "VirHostMatcher," and "PHP"), individual machine learning classifiers taking into consideration multiple top hits for each input virus were optimized as follows. A training set was built from the hits obtained from NCBI Virus RefSeq release 201 [52] against the iPHoP_db_Sept21 database, using similar cutoffs as for the benchmarks (see above) but considering for each input virus the 50 best hits (blast) or 30 best hits (all other methods). All hits were associated with the corresponding host genome representative (see "Virus–host training sets and host databases" above), and for each input virus, all host genome representatives with one hit were considered as a candidate host.

For each pair of input virus–candidate host, the different hits obtained for this virus were gathered as follows. First, the phylogenetic distance between the host genome representative of each hit and the candidate host was obtained from the GTDB-tk-generated trees (see above), so that hits can be ordered by distance to the candidate host. Next, depending on the tools, 1 to

3 scores were used to describe the strength of the hit, and all the hits for a given input virus are tallied, i.e., the number of hits observed for a given distance and set of scores is tabulated. The resulting matrices then serve as input to either neural network or random forest classifiers. For more detailed information about the cutoffs, score selection, and transformation used for each tool, please see S5 Fig.

For classifier training, a subset of 20,000 to 60,000 virus–host instances were randomly selected for each tool, with the following constraints: (i) between 60% to 85% of incorrect virus–host pairs, i.e., instances where the candidate host was assigned to a genus different from the host genus listed for this virus in the database, and (ii) between 45% to 70% of instances with a "known" virus, i.e., for which the virus had an AAI percentage of 70% or higher to the closest reference. These constraints were included to ensure that the training set was not too unbalanced in favor of (i) incorrect predictions, since most hits are to genomes from a different genus than the host, and (ii) "known" viruses, which typically represent the majority of databases and could bias the classifiers. A subset (10%) of these training data were set aside and used as a common validation set when comparing different versions of each classifier (see below). Further, for each instance, 3 different sets of hits were used: one including all the hits obtained for the virus, one including only a random subset (from 0% to 100%) of hits, and one including only one randomly selected hit with a distance ≤4 to the candidate host (if any) and all hits with a distance >4, or one randomly selected hit among all hits if all display a distance >4. This random subsampling of hits was included to simulate different levels of representation of host diversity in the database, since current bacteria and archaea genome databases do not provide an even coverage of the global diversity, and isolated viruses used here for training are likely to be biased towards well-represented hosts.

## Optimization and evaluation of single-tool classifiers for iPHoP

All dense and convolution networks were built using TensorFlow 2.7.0 [65], and all random forest classifiers were built with the Tensor Flow Decision Forests v0.2.1, both within the Keras 2.7.0 Python library [66]. Classifiers were trained on the corresponding training set, using 80% of the data for training and 20% for validation ("validation_split = 0.2" for the neural networks). The Adam optimizer was used to train all the neural networks. Classifier parameters including the number of layers, kernel size, and dilation rate for convolution networks, number of dense layers for dense networks, and number of trees and maximum tree depth for random forest classifiers were optimized for each individual classifier using the Optuna v2.5.0 framework [67], by running 100 training trials (see S5 Fig). For each type of classifier (convolution network, dense network, and random forest classifier), the 5 best versions based on minimum Binary Cross Entropy loss (for networks) or maximum accuracy (for random forests) on the common validation set (see above) were selected as potential candidates.

To select the optimal combination of classifiers, these candidates were then applied to the test set, and the results obtained on the nonambiguous cases were observed (i.e., blast hit ≥10 kb, CRISPR match with 0 mismatches, WIsH $p$-value $\leq 1 \times 10^{-5}$, VHM score ≥0.8, PHP score ≥1,450). For each classifier, the 10th percentile of scores for these nonambiguous cases where the classifier prediction was correct was used as an estimate of a "high-confidence" score for this classifier, and the number of incorrect predictions with a score higher than this cutoff was used as an estimate of the error rate, i.e., incorrect prediction with a score comparable to nonambiguous correct predictions. This error rate was then used to iteratively select classifiers by first selecting the one with the lower error rate, then selecting additional classifiers if they provided ≥5% additional correct prediction among nonambiguous cases, or if they "corrected" ≥10% of the previous false-positive errors. If no classifier fulfilled these conditions, the

selection process was stopped. Ultimately, all selected classifiers (see S4 Table) were run on the full test set to derive Precision-Recall curves and FDR estimations.

## Training, optimization, and evaluation of iPHoP main random forest classifier (combined-hosts-RF)

To integrate signal from multiple approaches, a random forest classifier ("**combined-hosts-RF**") was trained to obtain a single confidence score for a given virus–candidate host pair based on the score obtained for all individual classifiers selected (see S4 Table). Specifically, for each virus–candidate host pair used in the training set (see above), the following information were included for each selected classifier: the score obtained for the virus–candidate host pair, the rank of this pair among all candidate hosts considered for this given virus, and the difference between the score of the pair and the highest score obtained for the given virus. This led to a final input matrix with 30 columns, i.e., 3 features (score, rank, and distance to best score) for each of the 10 selected classifiers. A balanced training set was built from the training sets created for each individual classifier (see above), including 700 randomly sampled viruses with at least 1 blast hit and 1 CRISPR hit, 700 each from viruses with either at least 1 blast hit or 1 CRISPR hit, and 700 viruses with neither blast or CRISPR hits. For each selected virus, up to 10 correct and up to 5 incorrect predictions (i.e., candidate virus–host pairs) were randomly selected. Eventually, the balanced training set included 17,105 correct and 13,960 incorrect virus–candidate host pairs.

Random forest classifiers were built using the TensorFlow Decision Forests v0.2.1 [65] package within the Keras 2.7.0 python library [66], with parameters optimized with the Optuna v2.5.0 framework [67]. Parameters to be optimized included maximum tree depth (between 4 and 32), minimum number of examples in a node (between 2 and 10), and number of trees (between 100 and 1,000). A total of 100 trials were performed, each was evaluated on the test dataset, the 5 classifiers with the highest accuracy were selected as the best candidates, and the candidate with the highest recall at 5% FDR was then selected as the final combined-hosts-RF classifier.

## Integrating iPHoP classifiers and RaFAH into a final host prediction

In order to rank host predictions for individual viruses obtained with different methods, and since the scores from different classifiers are not directly comparable, the test dataset was used to transform raw scores into empirical FDRs. Specifically, the PPV, i.e., the number of correct predictions divided by the total number of predictions, which corresponds to 1 minus the FDR, was computed on sliding windows of each tool score from 0 to 1, with window size 0.05 (for Blast Conv_87, Blast RF_39, CRISPR Conv_85, CRISPR Dense_15) or 0.01 (for combined-hosts-RF, and RaFAH). For each tool, a generalized linear model was then fitted on these values using the mgcv v1.8–36 library [68] in R v 4.0.5 [69] with REML estimation, and an empirical PPV and FDR was then calculated for scores ranging from 0 to 1 by steps of 0.001.

These empirical PPVs are then used in the iPHoP framework to derive a single composite score for each virus–candidate host genus pair as follows. For each pair, all methods with PPV <0.5 are first discarded. Next, the method with the highest PPV, i.e., the lowest FDR, for this pair is selected as the source of the main FDR. To take into account prediction from other methods, which passed the PPV threshold, i.e., were ≥0.5, the FDRs from these additional predictions are then multiplied by 2 (to rescale between 0 and 1), and the final composite score is then calculated as 1 minus the product of the main FDR and the additional "rescaled" FDRs, if any. This means that additional methods pointing to the same virus–host genus pair can only

improve the composite score, as they will multiply the main FDR by factors always ≤1. Finally, a similar empirical approach based on the test dataset was used to transform these composite scores in PPVs (see above), and these empirically estimated PPVs are provided to iPHoP users as "Confidence score" in the result files. By default, only predictions with a confidence score ≥90, i.e., an estimated FDR <10%, are included in the summary output file; however, users can select any confidence score ranging from 75 to 100.

To enable this integration of results from host-based tools and RaFAH, the predictions from RaFAH had to be converted into GTDB-compatible taxa. To this end, each genus listed in the RaFAH output file was searched for in the GTDB metadata files, and the list of genomes associated with this RaFAH genus along with their GTDB genus-level taxon was tallied. Each RaFAH genus was then associated to all GTDB genus-level taxa representing ≥50% of the genome list if the list included <10 genomes, ≥20% of the genome list if the list included 10 to 100 genomes, or ≥10% if the list included ≥100 genomes. This approach provided GTDB genus-level taxa for 595 RaFAH genera, with 492 linked to a single taxa, and 90 linked to 2 taxa, often closely related (e.g., "Thioalkalivibrio" and "Thioalkalivibrio_B," "Pseudothermotoga_A," and "Pseudothermotoga_B").

## Benchmarks and comparison to other integrated host prediction approaches

Three other tools providing host prediction based on multiple approaches were benchmarked on the same test dataset (see above) as iPHoP. VirHostMatcher-Net (July 2021 version; [31]) was run on the test dataset with default parameters, requesting the top 100 predictions to be included in the output files, and using the default host database provided with the tool. Phis-Detector (February 2021 version; [30]) was run on the test dataset with the following parameters: "--min_mis_crispr 2 --min_cov_crispr 70 --min_per_prophage 30 --min_id_prophage 70 --min_cov_prophage 30 --min_PPI 1 --min_DDI 5 --min_per_blast 10 --min_id_blast 70 --min_cov_blast 10", and using the default database provided with the tool. Finally, Vir-Matcher v0.3.2 [29] was run on the test dataset via its KBase App [70]. Since no host database was provided with VirMatcher, a custom host genome database was built based on the RefSeq genomes that displayed at least 1 hit to any of the test dataset virus with blast, CRISPR, or WIsH.

For each tool, the prediction with the highest score was considered as the host genus predicted for a given virus, excluding predictions to hosts with unknown genera. These "best hit" predictions were then used to evaluate the recall of each tool, i.e., the number of correct host genus predictions, at different FDR level, either on the complete test dataset or when restricting to specific ranges of "novelty," i.e., AAI to the closest reference ranging from 0% to 5%, 5% to 10%, 10% to 30%, 30% to 60%, or 60% to 100%.

To evaluate the impact of sequence length and completeness on iPHoP performance, datasets of partial sequences were generated from the test set using an in-house Perl script. Specifically, 10 replicates of 500 random sequences were generated with the following length: 20 kb, 10 kb, 5 kb, and 1 kb, or completeness: 50%, 20%, 10%, and 5%. For reference, 10 random datasets of 500 complete genomes from the same test set were also generated. Host genus was predicted on these input sequences with iPHoP using default parameters, and recall and FDR were evaluated as described above.

To evaluate how eukaryotic virus sequences are handled by iPHoP, we performed host prediction with iPHoP using default parameters on 8,128 eukaryotic virus genomes obtained from NCBI RefSeq r214 [52]. These were selected based on the host information provided in the RefSeq database, and their taxonomy was also obtained from the same RefSeq database.

Finally, to evaluate the potential for false-positive host predictions when the real host genus was not represented in the host database, we performed host prediction with iPHoP on the test dataset using a series of filtered database from which individual genera were removed. We then compared the predictions obtained at 3 given cutoffs (75, 90, and 95) to the real host taxonomy, i.e., whether a prediction was provided, and whether the prediction corresponded to the same family, class, and/or order as the real host. For this benchmark, only host-based predictions were considered, as RaFAH predictions do not rely on the availability of a host genome.

## Evaluation of iPHoP host predictions on high-quality genomes from the IMG/VR database

To evaluate iPHoP on real metagenome-derived virus genomes, 216,015 high-quality genomes from the IMG/VR v3 database [8], i.e., metagenome-derived viral genomes estimated to be ≥90% complete based on CheckV v0.4.0 [71], were processed with iPHoP v1.0, using the iPHoP_db_Sept21 database. These sequences were previously identified as prokaryotic virus genomes, i.e., all viruses predicted to infect eukaryotes were excluded, by the IMG/VR v3 pipelines [8]. Host genus prediction was based on the host genus with the best iPHoP composite score for each input sequence, with a minimum score cutoff of 75. Metadata for IMG/VR sequences, including corresponding study and dataset if available, were obtained from the IMG/VR database (2020-10-12_5.1 version) [8]. Temperate and virulent phages were identified based on BACPHLIP v0.9.6 [55] with a minimum score of ≥0.8. Metadata for the host genome, including the corresponding study and dataset if available, were obtained from the IMG and Gold databases (information downloaded in January 2022; [35,72]). To verify that this dataset did not overlap significantly with the training and test sets, genome-wide AAI was computed between the 216,015 high-quality genomes and the genomes included in the training and test sets using the AAI estimation script provided with the Metagenomic Gut Virus catalogue (https://github.com/snayfach/MGV/blob/master/aai_cluster/README.md). Overall, 90% of all IMG/VR v3 sequences, and 87% of IMG/VR v3 sequences with at least 1 host predicted, displayed less than 10% AAI to their closest reference in the training or test sets, confirming that the IMG/VR sequences are meaningfully distinct from these training and test sets. VirHostMatcher-Net [31] was applied to the same dataset for comparison, using the same version, options, and cutoffs as previously described (see above).

A custom host database was built to evaluate the impact of bacterial and archaeal MAGs on host prediction by iPHoP. Specifically, all MAGs were excluded from the host database, i.e., the whole GEM dataset and all sequences in GTDB or IMG indicated as "metagenome-derived." Host prediction with iPHoP was then performed for the same IMG/VR sequences, using similar parameters as described above. Similarly, to further evaluate the impact of "local" MAGs, i.e., MAGs derived from the same study as the input sequences, on iPHoP prediction, 3 custom host databases were built for 3 collections of metagenomes (Gold Study Id Gs0110170, Gs0114290, and Gs0114820), in which the MAGs derived from these specific studies were removed. These host databases were then used for host prediction with iPHoP on all IMG/VR sequences assembled from these same studies. In both cases, the list of viruses with host prediction at different score cutoffs was compared to the list of viruses for which host prediction was available when using the whole host database.

To represent the diversity of hosts included in these IMG/VR-derived host predictions, the GTDB bacteria and archaea trees were plotted using the ggtree v2.4.1 package [73], with clades collapsed at the phylum level. Each phylum was then colored according to the status of its member genera, i.e., whether each host genus is associated with an isolated virus in RefSeq, a

host prediction with iPHoP score ≥95, a host prediction with iPHoP score ≥90, or no isolate or host prediction. To verify whether iPHoP host predictions linked viruses from each main biome to host taxa consistently found in the same biomes, the GEM dataset [36] was used to evaluate the biome distribution of individual host genera. Specifically, each GEM MAG was associated to its corresponding genus and original sample biome, if this information was available (*n* = 38,556). Each genus was then associated with a given biome if ≥50% of the corresponding MAGs originated from a sample of this biome (*n* = 3,500 genera) or was considered as "Detected across multiple biomes" if the majority biome represented <50% of the genus MAGs (*n* = 90 genera).

## Supporting information

**S1 Fig. Characteristics of the test dataset.** (**A**) Distribution of the host genera for the test dataset. Note: Only genera associated with ≥5 viruses are included, another 125 host genera were associated with <5 viruses and are not displayed. (**B**) Distribution of AAI to the closest reference in NCBI RefSeq for the test dataset. The corresponding list of viral genomes included in the test dataset is provided in S2 Table. Source data are available in S1 Data (Source data 1 and S2 Table).
(TIF)

**S2 Fig. Comparison of different host prediction approaches on different subsets of the test dataset.** Total number of predictions and number of correct predictions (y-axis) obtained at any rank for each tool (x-axis) on sequences from the test dataset (S2 Table). For each tool, the number of correct predictions is indicated by the colored bar, while the total number of predictions is indicated by the gray bar. The top panel displays the results obtained on the entire test dataset (*n* = 1,870). The middle panel includes results obtained for all phages predicted as temperate, either via BacPhlip or based on the genome annotation (*n* = 949). The middle panel includes results obtained for all phages predicted as virulent by BacPhlip (*n* = 663). Source data are available in S1 Data (Source data 1).
(TIF)

**S3 Fig. Characteristics of the high-quality IMG/VR genomes.** (**A**) Number of high-quality viral genomes from IMG/VR v3 identified across the 5 major biomes in the database. Genomes sampled from other biomes of lacking a biome information are gathered in the "Other" category. (**B**) Distribution of the average amino acid identity between IMG/VR v3 viral genomes and the NCBI Viral RefSeq v203. Source data are available in S1 Data (Source data 4).
(TIF)

**S4 Fig. Overlap between host-based tools for individual viruses.** For each host-based tool included in the benchmark (see Fig 1), the overlap in terms of input sequence for which a correct prediction was obtained is presented here as an upset plot. The intersection size represents the number of phages with correct prediction using the combination of methods indicated at the bottom. This number is also indicated above each bar, and the bar color indicates the number of tools included in the combination. Source data are available in S1 Data (Source data 1).
(TIF)

**S5 Fig. Schematic of the data transformation and classifier architectures used in iPHoP.** (**A**) Summary of the cutoff and metrics used for each host-based tool considered in iPHoP (see S1 Table). (**B**) Overview of the 3 different types of classifiers evaluated in iPHoP. The different parameters optimized using the Optuna framework are highlighted in blue. For varying numbers of layers, the same parameters were optimized for each layer, but each was optimized

separately, i.e., the parameters values were independent between the different layers.
(TIF)

**S6 Fig. ROC and Precision-Recall curves for single-tool classifiers.** For each host-based tool, the ROC curves (left) and Precision-Recall curves (right) based on the test dataset are presented for the 5 best classifiers of each type and compared to the "naive" approach, i.e., best hit based on the raw score. FPR, false positive rate; PPV, positive predictive value; TPR, true positive rate. The 1-to-1 line is indicated as a dashed black line on the ROC curves. Random forest classifiers were only evaluated for Blast and CRISPR approaches. Source data are available in S1 Data (Source data 5).
(TIF)

**S7 Fig. Percentage of correct host predictions obtained for viruses with different degrees of "novelty".** The number of correct host predictions was evaluated for 3 different score cutoffs corresponding to 20%, 10%, and 5% estimated FDR (false discovery rate). Input viruses were classified into 5 categories (x-axis) based on their AAI (average amino acid identity) to the closest reference phage genome. The number of correct host predictions is indicated for each iPHoP classifier (see Fig 3A) and for the composite score considering all classifiers ("Final iPHoP prediction"). Source data are available in S1 Data (Source data 3).
(TIF)

**S8 Fig. Comparison of different integrated host prediction tools, including iPHoP, on the test dataset.** Standard Receiver Operating Characteristic (left) and Precision Recall (middle) curves for the 4 integrated host prediction approaches compared. To take into account the number of predictions provided by each tool, a third plot (right panel) indicates the positive predictive value (y-axis) when considering an increasing number of predictions (x-axis). To obtain this, cutoffs were progressively lowered to include an increasing number of predictions for each tool and prioritize the highest confidence ones, i.e., starting with the highest PPV possible. For the ROC curve, a 1-to-1 line is indicated with a dashed black line. For the Precision Recall and PPV curves (middle and right panels), the red and purple dashed lines indicate 5% and 10% false discovery rates, respectively. Source data are available in S1 Data (Source data 3).
(TIF)

**S9 Fig. Type of host prediction obtained for high-quality IMG/VR v3 genomes with different degrees of "novelty".** High-quality genomes from the IMG/VR v3 database for which a host prediction was obtained with iPHoP (score $\geq$90) were binned based on the average amino acid identity (AAI) to the closest reference in NCBI RefSeq Virus r203 (x-axis). Predictions entirely based on host-based tools are indicated as "Host only," predictions exclusively based on RaFAH are indicated as "Phage only," and predictions where both types of tools were consistent and with iPHoP score $\geq$90 are listed as "Both." Source data are available in S1 Data (Source data 4).
(TIF)

**S10 Fig.  Breakdown of iPHoP host predictions for high-quality IMG/VR v3 genomes assigned as virulent (top) or temperate (bottom).** Similar as Fig 4A and 4B, the left panel shows the distribution of the best score provided by iPHoP for the corresponding subset of IMG/VR v3 quality genome (top: virulent, bottom: temperate), organized by ecosystem. For each vOTU, the best score from iPHoP was considered if $\geq$75, or the vOTU was considered as not having a predicted host. The right panel shows the type of signal used to achieve host prediction with a score $\geq$90. "Host-based" includes all 5 host-based tools, while "Phage-based"

includes predictions obtained with RaFAH. "Both" includes consistent predictions obtained with RaFAH and at least 1 host-based tool. Temperate and virulent phages were identified via BACPHLIP [55] with a minimum score of 0.8 and based on genome annotation (see Methods). Source data are available in S1 Data (Source data 4).
(TIF)

**S11 Fig. Number of species and iPHoP prediction per host genus.** Each dot represents a host genus with at least 2 species, with the x-axis reflecting the total number of species in the genus, and the y-axis reflecting the total number of IMG/VR v3 HQ sequences predicted to infect this host genus with an iPHoP score ≥ 90. Host genera and species were obtained from the GTDB database [34]. The right panel presents a zoomed-in version of the area highlighted with dashed black lines in the left panel. Source data are available in S1 Data (Source data 4).
(TIF)

**S12 Fig. Performance of iPHoP on partial genomes.** Partial genome assemblies were simulated by taking 500 random sequences from the testing set and selecting a random subset of fixed length (20 kb, 10 kb, 5 kb, and 1 kb) or fixed completeness (50%, 20%, 10%, and 5%), before processing with iPHoP for host genus prediction. This process was repeated 10 times, and the standard error across the 10 replicates is reported on each bar plot. Host prediction with iPHoP was also obtained for the same sets of 500 sequences from the testing set using complete genomes, for reference ("100% completeness," colored in grey on the figure). For each input type, the recall (i.e., number of sequences with a host prediction, top), and FDR (i.e., percentage of erroneous predictions among all predictions, bottom) is indicated for different minimum iPHoP score cutoffs (75, 90, and 95). Source data are available in S1 Data (Source data 6).
(TIF)

**S13 Fig. Performance of iPHoP when no reference genomes from the correct host genus is available in the host database.** "Novel" host genera, i.e., cases in which the correct host was not represented in the host database, were simulated by recomputing iPHoP host prediction using filtered databases where all genomes from a given genus were removed. For this benchmark, only host-based predictions were considered and RaFAH phage-based predictions were ignored, as the latter did not rely on the availability of reference genomes in the host database. The same test dataset was used as in the regular benchmarks (see S1 Table and S1 Fig). For each minimum score cutoff (75, 90, and 95), all phages for which a correct host genus prediction was obtained with the standard database at this given cutoff were considered, and the prediction with the corresponding filtered host database was compared to the real host taxonomy. The category "no prediction" correspond to cases where iPHoP did not provide a prediction at the selected cutoff when using the filtered host database. The default score cutoff (90) is highlighted with a star symbol. The score cutoff of 75 is the lowest possible minimum score available in iPHoP. Source data are available in S1 Data (Source data 7).
(TIF)

**S1 Table. List of individual tools benchmarked, included in, and/or compared to iPHoP.**
(XLSX)

**S2 Table. List of viral genomes included in the test dataset, obtained from NCBI GenBank, and used to evaluate the performance of individual and integrated tools.**
(XLSX)

**S3 Table. Number and percentage of correct host predictions for different individual tools at different host taxonomic ranks.** For each tool and rank, the number and percentage of

correct predictions is indicated for all available predictions, the top 50%, or the top 10%, ranked based on each tool's score (see Methods).
(XLSX)

**S4 Table. Characteristics of the single-tool classifiers considered for inclusion in iPHoP.** The classifiers eventually included in iPHoP v1.0 are indicated with a "x" symbol in the column "Classifiers selected for inclusion in iPHoP."
(XLSX)

**S5 Table. Overview of iPHoP host genus prediction for IMG/VR high-quality sequences across different host databases, with and without (local) MAGs.** The first tab includes the summarized results for the 2 benchmarks, i.e., a custom database without MAGs for all IMG/VR sequences, and custom databases without "local" MAGs (i.e., from the same study) for 3 specific studies. Additional tabs provide the list of IMG/VR sequences with a host prediction from the complete host database, but either no prediction or a lower iPHoP score prediction when using a "no MAG" or "no local MAG" host database.
(XLSX)

**S6 Table. Number of viruses predicted for each host genus.** Host genera with at least 2 species and considered in Fig 5 are listed in the first tab, while host genera composed of a single species are indicated in the second tab.
(XLSX)

**S7 Table. Erroneous host prediction by iPHoP for eukaryotic viruses.** Eukaryotic virus genomes were obtained from the NCBI RefSeq r214 database. For each virus with a prediction with iPHoP score $\geq 75$, the best predicted host (i.e., maximum score) is indicated, along with the method within iPHoP that yielded this prediction (see Fig 3A).
(XLSX)

**S1 Data. Underlying data for main and supplementary figures (Figs 1–5, S1–S4 and S6–S13).** The first tab ("summary") indicates which source data tab and/or supplementary table corresponds to which figure
(XLSX)

## Acknowledgments

This manuscript has been authored by an author at Lawrence Berkeley National Laboratory under Contract No. DE-AC02-05CH11231 with the US Department of Energy. The US Government retains, and the publisher, by accepting the article for publication, acknowledges that the US Government retains a nonexclusive, paid-up, irrevocable, worldwide license to publish or reproduce the published form of this manuscript, or allow others to do so, for US Government purposes.

## Author Contributions

**Conceptualization:** Simon Roux, Andrew Tritt.

**Data curation:** Simon Roux, Shareef M. Dabdoub.

**Formal analysis:** Simon Roux, Felipe H. Coutinho, Bas E. Dutilh, Stephen Nayfach, Andrew Tritt.

**Investigation:** Simon Roux, Antonio Pedro Camargo, Andrew Tritt.

**Methodology:** Simon Roux, Antonio Pedro Camargo, Stephen Nayfach, Andrew Tritt.

**Software:** Simon Roux, Antonio Pedro Camargo, Felipe H. Coutinho, Andrew Tritt.

**Validation:** Simon Roux, Antonio Pedro Camargo, Felipe H. Coutinho, Bas E. Dutilh, Stephen Nayfach.

**Visualization:** Simon Roux, Antonio Pedro Camargo, Shareef M. Dabdoub, Bas E. Dutilh, Stephen Nayfach.

**Writing – original draft:** Simon Roux, Antonio Pedro Camargo, Felipe H. Coutinho, Shareef M. Dabdoub, Bas E. Dutilh, Stephen Nayfach, Andrew Tritt.

**Writing – review & editing:** Simon Roux, Antonio Pedro Camargo, Felipe H. Coutinho, Shareef M. Dabdoub, Bas E. Dutilh, Stephen Nayfach, Andrew Tritt.

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
