## [Editor Report · Decision Letter 0]

5 Sep 2022

Dear Dr. Roux, 

Thank you for submitting your manuscript entitled "iPHoP: an integrated machine-learning framework to maximize host prediction for metagenome-assembled virus genomes" for consideration as a Methods and Resources by PLOS Biology.

Your manuscript has now been evaluated by the PLOS Biology editorial staff and I am writing to let you know that we would like to send your submission out for external peer review.

Once your full submission is complete, your paper will undergo a series of checks in preparation for peer review. After your manuscript has passed the checks it will be sent out for review. To provide the metadata for your submission, please Login to Editorial Manager (https://www.editorialmanager.com/pbiology) within two working days, i.e. by Sep 07 2022 11:59PM.

Kind regards,

Paula

Senior Editor

PLOS Biology

---

## [Decision Letter · Decision Letter 1]

14 Nov 2022

Dear Dr. Roux,

Thank you for your patience while your manuscript "iPHoP: an integrated machine-learning framework to maximize host prediction for metagenome-assembled virus genomes" was peer-reviewed at PLOS Biology. It has now been evaluated by the PLOS Biology editors, an Academic Editor with relevant expertise, and by several independent reviewers. 

In light of the reviews, which you will find at the end of this email, we would like to invite you to revise the work to thoroughly address the reviewers' reports.

As you will see below, the reviewers agree that there are issues regarding benchmarking and evaluation of the method. After discussing with the Academic Editor, we think that the validation on metagenomic data is not necessary for publication. This is because benchmarking is not possible without having host information for a specific phage, as it happens with phage genomes in metagenomic data. We think that benchmarking using phage genomes with known host information and then apply to a large set of metagenome-derived viral genomes is acceptable here. We do think it is important that you clarify the workflow, as both reviewers point out. You should clearly show how you handled the possibility of information leak from training to test data, as highlighted by reviewer #1. You should also discuss the comment from reviewer #2 on "correct" and "incorrect" predictions along with your approach to reduce false positives taking into account the fact that there is no guarantee that the host genome is present in the host-db. E.g., what would happen if all members of the host genus are removed from the input? Would the method predict some random genus, a closely related genus, or nothing? We think that would be a more realistic evaluation of false positives in the metagenomic context.

Given the extent of revision needed, we cannot make a decision about publication until we have seen the revised manuscript and your response to the reviewers' comments. Your revised manuscript is likely to be sent for further evaluation by all or a subset of the reviewers.

**IMPORTANT - SUBMITTING YOUR REVISION**

*Re-submission Checklist*

*Published Peer Review*

*PLOS Data Policy*

*Blot and Gel Data Policy*

Sincerely,

Paula

---

Senior Editor

PLOS Biology

REVIEWS:

Reviewer #1: Machine learning, genomics.

Reviewer #2: Microbiome discovery.

Reviewer #1: This is an integrated, multistep approach to adjudicate viral genomes from metagenome datasets to their putative host. The work is well executed, and the draft is carefully written. Below, there are some comments that aim at pressure testing some of the central statements. 

Title: The title refers to "metagenome assembled virus genomes" - In reality it should refer to phage viral genomes

Abstract: The manuscript carefully refers to host genus predictions. The abstract should be more precise - currently it refers to "host predictions" indicating "reliable host predictions at the genus rank" would be adequate.

On lines below 246: There is an important discussion on the presence of cognate host sequence in the material from where the metagenome was sequenced ("local"). Certainly, co-occurrence of the phage and the putative genome is a valuable input. Overall, a substantial (>25%) proportion of MAGs used for host predictions were obtained from metagenomes generated in the same study from which the input phage was derived. It is unclear how this information was used to evaluate performance of iPHoP. There is an analysis described below line 269, but it is unclear that it reaches the value of developing models that are "local" aware. The authors are fully aware of the potential ("increasing the diversity of isolated phage-host pairs from various environments will likely help refine these predictions in the future") but is not clear that they have used available data to that goal.

One of the critical aspects of machine learning approaches is the management of "leaking" from training sets to test sets. Here, there are a number of approaches aiming at controlling for such bias. However, it is unclear how for example IMG/VR phage sequences and pairs were checked for absence of the datasets used during training.

Figure 1 shows the comparison of different host prediction approaches on a single test dataset. It is placed early in the text, before iPHoP is presented.The issue with this mode of drafting the manuscript is that there is no formal comparative of the new tool here. It is true that there are a number of comparative plots later in the manuscript but are more selective. 

More generally, it is difficult to assess in a first read how much of the tool goes beyond integrating existing tools, using ensemble of machine learning approaches (eg., Figure 2), or using composite scores (eg., Figure 3), and when and how the taxonomy aware improvements are included. The work would benefit from an end-to-end view of the process of iPHoP. And as indicated above, clarity on what is used as truth.

Reviewer #2: The authors present iPHoP, a computational framework to associate metagenome-derived viral genomes with microbial hosts. Existing methods leverage both alignment-based and alignment-free tools to predict phage hosts, though these approaches suffer from limited taxonomic resolution and / or high false-discovery rates. iPHoP refines the output of such tools to yield genus-level associations that are both sensitive and specific, giving insight into the putative hosts of phage, including "novel" phage with low sequence similarity to phage with known hosts. When applied to metagenome-derived phage genomes the IMG/VR database, iPHoP increases the proportion of phage with host assignments by more than 10-fold for certain ecosystems. The authors also create a database building module, Bioconda recipe and a Docker container for users to apply iPHoP to their own datasets. 

Major Comments

Since the main usage of this software will be on metagenomics datasets, it would be great to have the authors benchmark and spend more time on those types of datasets. The authors use MAGs from a sample to predict the virus-host associations of viruses within that same sample (section starting 255). Which tool was used in the first step of IPHoP? Since so many of the associations (~30%) were from the MAGs alone, it seemed like there should be more benchmarking of this pipeline. For one, it relies on using BACPHLIP as a phage-finding method. Why was this picked and how accurate precise is it? Importantly, how do the other host-annotation prediction algorithms do on this dataset? How much do their predictions overlap? What would have been the results without the input of the additional MAGs?

The authors mention RaFah as having good overall performance. Since iPHoP runs as a refinement tool, why did the authors not apply it to RaFah as well? The authors show that it can improve the predictions of host-based methods (Figure 3). 

Figure S7 nicely shows an improvement over RaFah, but what host prediction tool was used in the primary step here? Since the percentage of prophage correctly assigned is highest for the combined and in almost all cases, results in a more than 10% increase, perhaps the prescription is to run all of these methods or at least some combo of them? Additionally, could the outputs of multiple tools be used as input for iPHoP? 

Concerning Figure 1, "correct" host predictions are assessed at the genus rank (line 105). It would be informative to have more taxonomic categories for Figure 1A than "correct" and "incorrect" predictions. For example, predictions may be "correct" at the ranks of family or class, and this information would be beneficial to understanding which existing tools are "close to correct" and which are way off the mark, even if the rest of the manuscript only considers predictions at the genus level. 

iPHoP seems to be able to predict multiple hosts, if they exist (for example, what Fig 2B shows). Can the authors expand the performance of various callers and their own as a function of number of hosts? 

Minor comments: 

Figure 5A is difficult to interpret given the different phyla shapes. Could the authors include numbers or a log-scale bar graph and/or color scale with the different categories for phage association? 

Why the distinction for genera with at least 2 host species? (line 258)

Supplementary Table 1 is useful. 

In Fig 1C, there is apparent bias for closely related phages when predicting hosts with alignment-free phage-based tools (HostPhinder). Why is assigning hosts by phage-phage k-mer similarity apparently dependent on AAI%, whereas assigning hosts by host-phage k-mer similarity is more-or-less agnostic to the level of phage novelty?

---

## [Decision Letter · Decision Letter 2]

23 Feb 2023

Dear Dr. Roux,

Thank you for your patience while we considered your revised manuscript "iPHoP: an integrated machine-learning framework to maximize host prediction for metagenome-derived viruses of archaea and bacteria" for consideration as a Methods and Resources at PLOS Biology. Your revised study has now been evaluated by the PLOS Biology editors, the Academic Editor, and one of the original reviewers.

In light of the reviews, which you will find at the end of this email, we are pleased to offer you the opportunity to address the remaining points from the academic editor in a revision that we anticipate should not take you very long. We will then assess your revised manuscript and your response to the comments from Academic Editor aiming to avoid further rounds of peer-review.

The remaining point refers to false positives, you should discuss your approach to reduce false positives taking into account the fact that there is no guarantee that the host genome is present in the host-db. E.g., what would happen if all members of the host genus are removed from the input? Would the method predict some random genus, a closely related genus, or nothing? We think that would be a more realistic evaluation of false positives in the metagenomic context.

Please also address the following editorial and formatting points:

1. DATA POLICY:

A) Supplementary files (e.g., excel). Please ensure that all data files are uploaded as 'Supporting Information' and are invariably referred to (in the manuscript, figure legends, and the Description field when uploading your files) using the following format verbatim: S1 Data, S2 Data, etc. Multiple panels of a single or even several figures can be included as multiple sheets in one excel file that is saved using exactly the following convention: S1_Data.xlsx (using an underscore).

B) Deposition in a publicly available repository. Please also provide the accession code or a reviewer link so that we may view your data before publication.

Regardless of the method selected, please ensure that you provide the individual numerical values that underlie the summary data displayed in the following figure panels as they are essential for readers to assess your analysis and to reproduce it: Figures 1ABC, 2CD, 3BC, 4, 5AB, and Supplementary Figures S1, S2, S3, S4, S6, S7, S8, S9, S10, S11, S12.

**Please also ensure that figure legends in your manuscript include information on where the underlying data can be found, and ensure your supplemental data file/s has a legend.**

**IMPORTANT - SUBMITTING YOUR REVISION**

*Resubmission Checklist*

*Published Peer Review*

*PLOS Data Policy*

*Blot and Gel Data Policy*

Sincerely,

Paula

---

Senior Editor

PLOS Biology

REVIEWS:

Reviewer #1: Amalio Telenti

Reviewer #1: Thanks for the careful revision of the manuscript

---

## [Editor Report · Decision Letter 3]

15 Mar 2023

Dear Dr Roux,

Thank you for the submission of your revised Methods and Resources "iPHoP: an integrated machine-learning framework to maximize host prediction for metagenome-derived viruses of archaea and bacteria" for publication in PLOS Biology. On behalf of my colleagues and the Academic Editor, Manimozhiyan Arumugam, I am pleased to say that we can in principle accept your manuscript for publication, provided you address any remaining formatting and reporting issues. These will be detailed in an email you should receive within 2-3 business days from our colleagues in the journal operations team; no action is required from you until then. Among those requests you will see that you will be asked to tone down the statement regarding the false-discovery rate in the abstract. In particular, it says in the abstract that the false-discovery rate is <10%, however taking into account the new benchmark the false-discovery rate would be higher. Please also add where the underlying data for the figures can be found in the Supplementary Figure legends. Please note that we will not be able to formally accept your manuscript and schedule it for publication until you have completed any requested changes.

PRESS

Sincerely, 

Paula 

---

Senior Editor

PLOS Biology
